# The Effectiveness of Frenotomy on Speech in Adults

Anna Lichnowska * 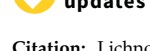 and Marcin Kozakiewicz 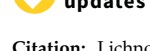

Department of Maxillofacial Surgery, Medical University of Lodz, 113th S. Żeromskiego, 90-549 Lodz, Poland; marcin.kozakiewicz@umed.lodz.pl
* Correspondence: anna.lichnowska@gmail.com

**Featured Application: The impact of tongue frenulum status in malocclusion is neglected in adults. Orthodontic and/or orthognathic treatment leads to a dental and visual correction of the face but leaves a functional deficiency in the form of a speech disorder. This study highlights the important functional role of the tongue frenulum not only in children but also in adult patients. Evaluation and correction of ankyloglossia should be part of the team treatment of malocclusion and facial skeletal deformities.**

**Abstract:** There is no publication concerning tongue-tie (TT) in adults, surprisingly. It is generally known that TT is mainly diagnosed in newborns and infants. It seems unlikely that TT does not cause functional disorders in adults, especially considering that TT has been present in organism since childhood. Thus, there is insufficient information about the influence of TT on adults′ speech production. The purpose of this study was the functional evaluation of lingual frenotomy on tongue mobility and speech in the adult Polish population. Methods: Methods were based on visual observation and examination of the oral cavity accompanied by visual and auditory examination of articulation. $X^2$ test, Kruskal–Wallis, analysis of variance (ANOVA), and Student′s t-test were used for statistical analyses. Conclusions: Tongue-tie is a serious condition in adults. Implementing surgical procedures to treat it improves the tongue′s mobility in every direction and improves speech clarity. The frenotomy should be implemented in patients suffering from malocclusion because the equilibrium of all the face muscles and skeleton is often disturbed and may lead to unstable functional effects of orthodontic and orthognathic treatment.

**Keywords:** tongue-tie; ankyloglossia; frenotomy; speech disorders; tongue mobility; adults; malocclusion

## 1. Introduction

Tongue-tie (TT) or ankyloglossia represents a condition in which the tongue′s movements are limited due to the lingual frenulum′s length. If the frenulum is too short, it tethers the tongue. Lingual frenotomy is a surgical procedure that should be implemented in most cases. Searching through medical databases such as PubMed, it is surprising that there is no publication concerning TT in adults. It is generally known that ankyloglossia is mainly diagnosed in newborns and infants. In this field, there are several publications and grading scales such as 1993 Hazelbaker′s [1], 1999 Kotlow′s [2], or 2009 Corylloss′ [3] classification of ankyloglossia in children [4]. Moreover, there are detailed descriptions of the prior and aftercare of patients.

Besides, there is insufficient information about TT′s influence on adults′ speech production and generally oral functions. It seems unlikely that ankyloglossia does not cause functional disorders in adults, especially considering that this form of ankyloglossia has been present in adults since childhood. Little is known about the anatomy, and even in anatomy textbooks designed for medical universities, the lingual frenulum is only briefly described. It is not easy to find articles giving full consideration of the frenum′s anatomical

structure, relationship to the floor of the mouth, and detailed descriptions of surgical procedures in adults [5].

As mentioned before, tongue-tie is generally diagnosed in children. Most classifications and anatomical descriptions consider the frenulum correct when it is visible or palpable [6] and does not limit the tongue's mobility [7]. Most classifications such as Kotlow's, Hazelbaker's, or Corryllos' use only a single feature of lingual frenum's visual appearance and categorize the tongue-tie grades (Table 1).

**Table 1.** Tongue-tie classifications according to Kotlow [2] and to Coryllos [3].

| Kotlow's | | Corryllos' | |
|---|---|---|---|
| Class I | Mild ankyloglossia 12 to 16 mm of the free tongue | Type I | Thin and elastic frenulum; attaches the tip of the tongue to the alveolar ridge, and the tongue forms a heart shape |
| Class II | Moderate ankyloglossia 8 to 11 mm of the free tongue | Type II | Thin and elastic frenulum; 2–4 mm behind the tongue tip, attaches by the alveolar ridge |
| Class III | Severe ankyloglossia 3 to 7 mm of the free tongue | Type III | Thick, fibrous and non-elastic frenulum; attaches mid-tongue to the floor of the mouth |
| Class IV | Complete ankyloglossia less than 3 mm of the free tongue | Type IV | The frenulum is not seen, but felt, with a fibrous or submucosal thick and shiny attachment from the base of the tongue to the floor of the mouth |

To briefly introduce classification, it is essential to describe the primary grades. The anterior tongue-tie is an attachment placed close to the tip of the tongue, while the posterior tongue-tie describes a frenum with a ventral tongue attachment; it is impossible to see it, as it is submucosal and needs to be palpated. What is more, in posterior TT, there is no visible tension [8]. In Kotlow's opinion, classes III and IV should be given special care as they severely restrict tongue range of movements. Kotlow also shares his criteria of the normal frenulum and its functionality as follows: the tip of the tongue should be able to protrude outside the mouth; without clefting, the tip of the tongue should be able to sweep the upper and lower lips easily; without straining, when the tongue is retruded, it should not blanch the lingual tissues to the anterior teeth, and the lingual frenum should not create a diastema between the mandibular central incisors [9].

On the other hand, Hazelbaker's assessment considers the appearance of the tongue accompanied by functions. However, this tool's great weakness in comparison with Kotlow's or Corryllos' is that it was designed for newborns and infants, and it is tough to transform this kind of assessment into adults. The multitude of classifications may create a dilemma when the frenulum should be considered normal or abnormal. Due to this fact, during the last decade, there has been a high increase in diagnosing TT worldwide [10]. Moreover, TT significantly impacts primary functions such as mastication, swallowing patterns, speech, and malocclusion, which should be carefully examined and considered before deciding frenotomy.

Speech impairments and articulation disorders are observed frequently in children with TT. Interdental lisping, rhotacism, multiple dyslalias [11], and any speech delay variation are mainly linked to humans' anatomical defects, altogether with weak mobility and motor skills [12]. Similar disorders can be observed in adults. The worldwide higher percentage of misarticulation is detected in rhotacism, and misarticulation of /R/ is mainly caused by a too-short frenulum in almost every language. It is easier to diagnose misarticulation in adulthood, which is caused by the frenulum and abnormal biting, mastication, and deglutition.

This study aimed to assess the functional evaluation of lingual frenotomy on tongue mobility and speech in the adult Polish population.

## 2. Materials and Methods

The presented study focused on the description of the impact of tongue-tie on primary functions, tongue's range of motion, and speech in the population of Polish adults, who also suffer from different severity of the malocclusion, and was approved by bioethical committee RNN/73/19/KE.

The research group consisted of 75 patients, Caucasian race, male ($n = 28$) and females ($n = 47$), with an average age of 24. Inclusion criteria: tongue-tie of any degree, Class II and III malocclusions, age > 18, no previous surgical treatment of lingual frenulum, and generally healthy. Exclusion criteria: patients did not sign the informed consent, any previous surgical treatment of lingual frenulum, lack of Class II and III malocclusion, or the patient did not obey the rehabilitation protocol (did not attend examinations or exercise according to recommendations).

Examiners were as follows: speech therapist, who conducted the full TT and logopaedics diagnostics, and maxillofacial surgeons, who conducted frenotomy. All of the patients were diagnosed with TT of different severity according to Kotlow's classification: 26 patients were diagnosed with mild ankyloglossia, 24 with moderate, and 25 with severe, and there were none with complete ankyloglossia. Out of 75 patients, 40 underwent a frenotomy. The surgical procedure was a horizontal incision, using a scalpel, without sewing, with local anesthesia by 4% Articain + Adrenaline. All of the patients qualified for TT's surgical treatment were doing some exercises before and after the surgery to gain the best possible mobility of the tongue, surrounding tissues, and muscles. Those exercises included: gentle massage of the mouth's floor along with the frenulum itself (using the index finger), horizontal and vertical movements of the apex, preadorsum, mediodorsum, and postdorsum. Vertical movements of the mediodorsum are movements in which the middle part of the tongue moves upward to the palate, and the tip is placed close to the upper central incisors. Postdorsum movements are performed while coughing and moving the soft palate. Due to such movements, patients could identify the tongue's location and particular muscles after the frenotomy. Besides, they were instructed to open their mouth wide so as not to feel any discomfort. Patients were asked to do some basic physical activities of the whole body like stretches facing down and neck rotations, accompanied by the massage of glottal and subglottal muscles.

During the essential intraoral examination, the assessment included: frenulum length according to Kotlow's classification, tongue shape, the look of the dorsum surface and its resting position, type of occlusion, teeth position, missing teeth, crowding, and other dental disorders, the shape of dental arches, palate, and lips, breathing, swallowing, biting, and chewing types, and pace of speech and its fluency. Breathing was observed during the whole examination, patients were also asked to do some exercises like squats, and it was checked whether the lips are sealed during physical activity and how they deal with inhales and exhales during the speech. To examine swallowing, biting, and chewing, patients were asked to drink water and eat an apple. Factors determining the proper function of swallowing were no or minimal activity of the mimic muscles and proper tongue's position (upward and backward movement of the tongue). For chewing, it was essential to observe jaws and lips' movement during the action and the lips' seal. When examining biting, it was crucial to check if the patients use their central incisors or is the act of biting moved laterally. Also, during the intraoral examination, the speech therapist looked at the tonsils and using the Pirquet's scale, assessed the size of tonsils. The motor assessment of tongue and lips was based on the adaptation of the Speech Organ Fitness Questionnaire [13], in which 20 tests were provided to assess both tongue and lips' motor skills. Table 2 presents the tests' descriptions. Each trial was rated on a scale from 0 to 3 points, where 0 indicated impossible movement and 3 indicated perfect movement. The next step was to conduct

a phonetic analysis. Each phoneme was assessed in 3 various voice word positions, i.e., front, middle, and back.

**Table 2.** Tongue and lip tests performed.

| Tongue Tests | Lip Tests |
|---|---|
| 1. Put out towards chin | 1. Pursing and stretching |
| 2. Put out towards nose | 2. Pursed lips lateral movements |
| 3. Lateral movements to lips' corners | 3. Smacking |
| 4. Put out wide on lips | 4. Lips shoot |
| 5. Corners upward movement | 5. Lips vibration |
| 6. Mediodorsum upward movement | 6. Circle shape lips |
| 7. Inside backward (to throat) movement | 7. Wide smile |
| 8. Lips licking | 8. Lips and teeth catch (up/down) |
| 9. Sound making | 9. Placing upper lip on lower and lower on upper |
| 10. Vertical–horizontal position (on the palate) | 10. Making fish mouth (simultaneous movement of upper and lower lip) |

During the examination, both auditory and visual analyses were used to determine the correct positioning of the speech organs, i.e., the appropriate speech organ position for the given phoneme accepted as a phonetics norm. The pronunciation of a particular phoneme was in agreement with the articulation place and was taken as the primary factor conditioning the assessment as correct or incorrect. A 5-grade scale of pronunciation was established: 1 point meant phoneme omission, 2 points meant a completely deformed phoneme, pronounced in an interdental manner, 3 points meant that the phoneme was pronounced too anteriorly, 4 points meant a phoneme was pronounced with a lateral position, and 5 points meant that the phoneme was pronounced correctly. A qualified and experienced speech pathologist conducted the pronunciation assessment.

After the logopedics diagnosis before the surgical treatment, patients were instructed how to stretch and relax the mouth floor, together with the subglottal and hyoid muscles. Moreover, they were asked to exercise the tongue's possible vertical movements and massage the frenulum and the floor of the mouth to prepare the tissues for the frenotomy. The patients exercised 15 min a day for three weeks before the surgery and six weeks after the frenotomy. The patients were instructed to stretch the wound, massage, and exercise since the first day after the frenotomy was completed to get the best possible range of movements and avoid the appearance of scar tissue, which could cause tongue-tie and mobility restriction again.

Statistical analysis was performed in Statgraphics Centurion 18 (Statgraphics Technologies Inc. The Plains City, US). The Kruskal–Wallis test was performed due to the lack of normal distribution or revealed a significant difference between pre-surgical versus post-surgical data variance. Categorical variables were tested by the $\chi2$ independence test. A *p*-value of less than 0.05 was considered statistically significant.

### 3. Results

The research results focused on main factors such as the movement range of the tongue, articulation disorders concerning worldwide known phonemes accompanied by some local phonemes, vertical–horizontal position of the tongue (resting positing), and primary function as follows: swallowing, mastication, and breathing.

Due to the prevalence of /t/, /d/, /n/, /r/, /l/ in the worldwide publications (international phonemes), sounds were evaluated before and after frenotomy (Figure 1). Moreover, it was checked whether the quality of the pronunciation of other sounds /ś/, /ź/, /ć/, /dź/, /f/, /w/, /s/, /z/, /c/, /dz/, /sz/, /ż/, /cz/, /dż/ (local phonemes) depends on frenotomy. Also, to complete the examination, a summary result for all nineteen of the examined sounds was done (all tested phonemes).

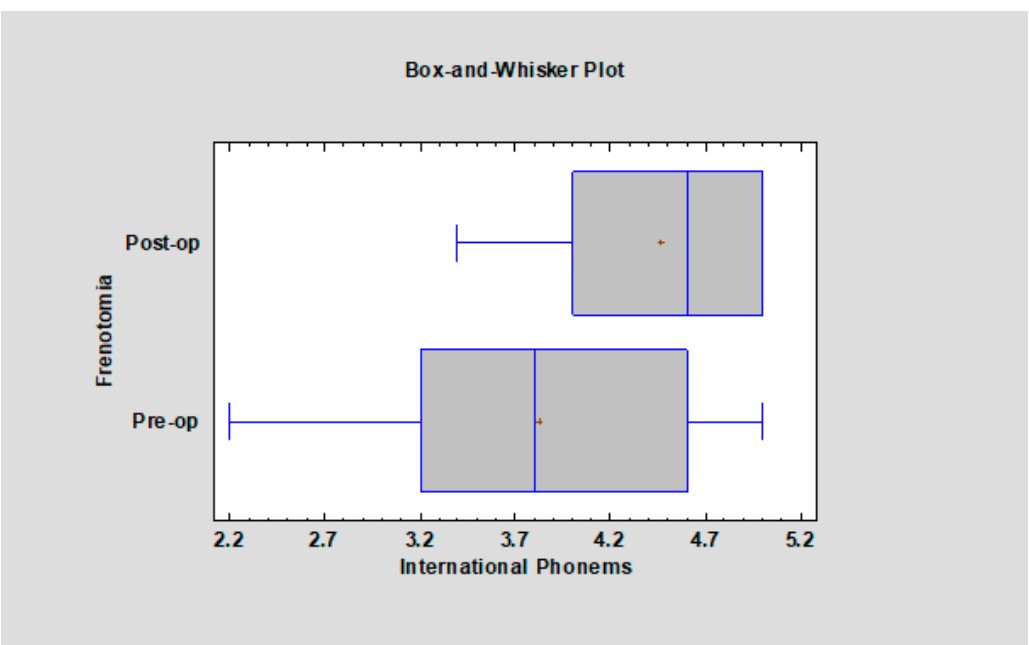

**Figure 1.** Frenotomy influence on international phonemes. Kruskal–Wallis test (test statistic = 10.28; $p < 0.005$).

It is clear that before frenotomy, more patients had difficulties in pronouncing sounds such as /t/, /d/, /n/, /r/, /l/, scoring approximately 3.7 points after conducting frenotomy, accompanied by some speech therapy patients who got better results and scored 4.3 points ($p < 0.05$). There is a significant difference between the quality of pronunciation before and after frenotomy (Figure 2).

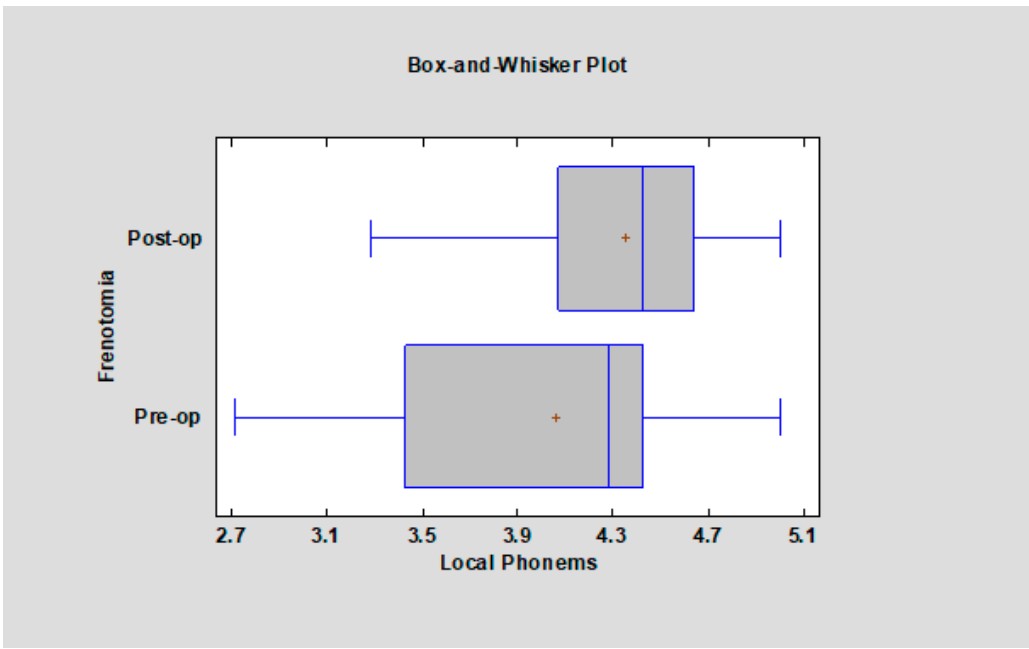

**Figure 2.** Frenotomy influence on local phonemes. Kruskal–Wallis test (test statistic = 3.40; $p = 0.06$).

As all of the local phonemes consist of one group here, and there was no significant difference between the pronunciation of investigated phonemes before and after frenotomy ($p = 0.06$). Also, comparing nineteen phonemes shows that there is a significant pronunciation improvement. In general (Figure 3), all patients who underwent frenotomy had

fewer articulation disorders after the frenotomy than those who did not have this surgical procedure performed ($p < 0.01$).

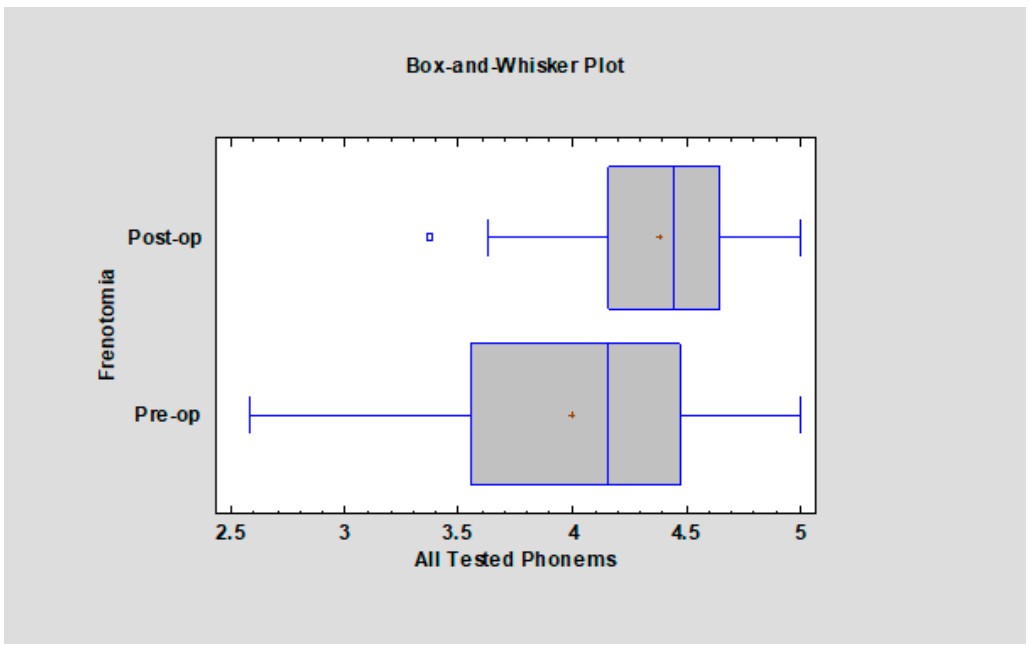

**Figure 3.** Frenotomy influence on all phonemes tested. Kruskal–Wallis test (test statistic = 7.43; $p < 0.01$).

It was essential to prove that Polish phonemes are very prone to any anatomical changes in the oral cavity or the maxillofacial skeleton. It was possible to find a relation (Figure 4) between laterogenia and too-short frenulum ($p < 0.05$).

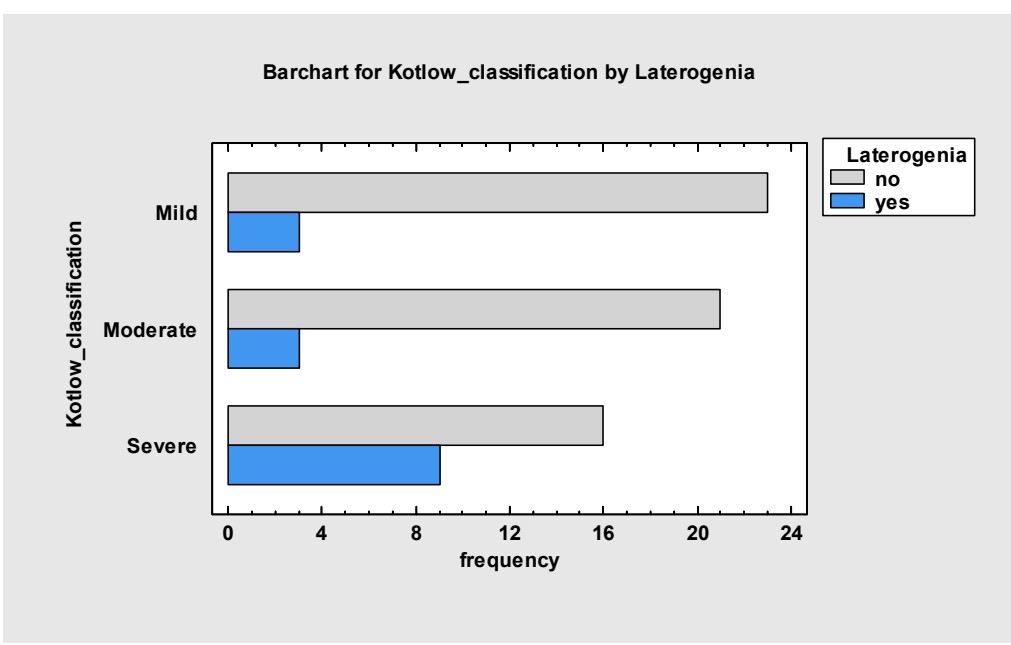

**Figure 4.** Relation of incorrect lingual frenulum to skeletal deformation ($\chi^2$ test 6.00; $p < 0.04$).

Tables 3 and 4 present the obtained results. It is observed that the mobility of the tongue significantly improved after surgery. That has a powerful impact on the quality of the articulated phonemes by patients too.

**Table 3.** Effect of frenotomy on tongue mobility.

| Tongue Test | Before Frenotomy | After Frenotomy | Significance |
|---|---|---|---|
| Tongue up | 1.46 ± 1.12 | 2.50 ± 0.55 | *p* < 0.0001 |
| Tongue down | 1.79 ± 0.60 | 2.31 ± 0.79 | *p* < 0.004 |
| Tongue lips corner | 2.83 ± 0.38 | 2.98 ± 0.16 | *p* < 0.05 |
| Wide tongue in out | 2.26 ± 0.78 | 2.88 ± 0.33 | *p* < 0.0001 |
| Spoon tongue | 2.03 ± 0.98 | 2.7 ± 0.52 | *p* < 0.005 |
| Arched tongue | 1.23 ± 1.21 | 2.43 ± 0.64 | *p* < 0.0001 |
| Tongue back | 2.37 ± 0.77 | 2.88 ± 0.33 | *p* < 0.0001 |
| Lips lick | 2.51 ± 0.66 | 2.9 ± 0.30 | *p* < 0.0005 |
| Tongue sound | 2.46 ± 0.82 | 2.83 ± 0.45 | *p* < 0.05 |
| Vertical–horizontal position | 1.31 ± 1.13 | 2.4 ± 0.67 | *p* < 0.0001 |
| Total tongue efficiency | 20.8 ± 4.59 | 27.3 ± 2.62 | *p* < 0.0001 |

Kruskal–Wallis test performed.

**Table 4.** Effect of frenotomy on the pronounced sounds.

| Phoneme Tested | Before Frenotomy | After Frenotomy | Significance |
|---|---|---|---|
| /ś/ | 4.28 ± 0.91 | 4.75 ± 0.63 | *p* < 0.01 |
| /ć/ | 4.25 ± 0.95 | 4.78 ± 0.48 | *p* < 0.01 |
| /ź/ | 4.18 ± 1.01 | 4.75 ± 0.54 | *p* < 0.005 |
| /dź/ | 4.23 ± 0.97 | 4.78 ± 0.48 | *p* < 0.01 |
| /s/ | 3.30 ± 0.99 | 3.31 ± 1.00 | n.s. |
| /z/ | 3.35 ± 0.95 | 3.36 ± 0.96 | n.s. |
| /c/ | 3.35 ± 0.95 | 3.36 ± 0.96 | n.s. |
| /dz/ | 3.33 ± 0.97 | 3.33 ± 0.98 | n.s. |
| /sz/ | 4.28 ± 0.85 | 4.75 ± 0.49 | *p* < 0.01 |
| /ż/ | 4.30 ± 0.88 | 4.75 ± 0.49 | *p* < 0.05 |
| /cz/ | 4.43 ± 0.87 | 4.83 ± 0.45 | *p* < 0.05 |
| /dż/ | 4.43 ± 0.87 | 4.83 ± 0.45 | *p* < 0.05 |
| /r/ | 3.95 ± 1.58 | 4.68 ± 0.57 | n.s. |
| /l/ | 4.63 ± 0.74 | 4.85 ± 0.43 | n.s. |
| /f/ | 4.55 ± 0.85 | 4.62 ± 0.75 | n.s. |
| /w/ | 4.60 ± 0.74 | 4.62 ± 0.75 | n.s. |
| /t/ | 3.48 ± 1.28 | 4.30 ± 0.82 | *p* < 0.005 |
| /d/ | 3.53 ± 1.24 | 4.28 ± 0.78 | *p* < 0.01 |
| /n/ | 3.55 ± 1.20 | 4.25 ± 0.84 | *p* < 0.01 |
| All tested phonemes | 4.00 ± 0.63 | 4.38 ± 0.40 | *p* < 0.01 |

Kruskal–Wallis test performed. n.s.—not statistically significant.

According to tonsils, the study did not prove that overgrown tonsils may lead to TT and, by this, to the tongue's restricted mobility. The χ2 result was 1.72, and the *p* = 0.42. Moreover, there is no relationship to incorrect breathing, which the TT might cause. The χ2 result was 0.42 and the *p* = 0.5. Results for chewing are as follows: the χ2 result was 0.71 and the *p* = 0.3. Incorrect biting is also not related to TT. The χ2 result was 0.025, and the *p* = 0.8. Swallowing patterns were also not related to TT, as the χ2 result was 3.39 and the *p* = 0.06.

The comparison on pronunciation before and after frenotomy showed a difference in articulation of dental phonemes, especially those which require a close contact with the hard palate /sz/, /ż/, /cz/, /dż/, /t/, /d/, /n/. The most significant changes happened to patients who were originally diagnosed with severe and moderate classes of TT, according to Kotlow [2]. They changed in such a pattern: severe–moderate, moderate–mild. As the changes occur, the pronunciation of the phonemes mentioned above improved significantly. Table 5 presents the obtained results for all nineteen phonemes.

**Table 5.** Pronunciation improvement of phonemes according to the severity of ankyloglossia (according to Kotlow). Positive values indicate improvement, '0' indicates no improvement, and negative values indicate deterioration.

| Phoneme Tested | Improvement Mild Class | Improvement Moderate Class | Improvement Severe Class | Significance $p < 0.05$ |
|---|---|---|---|---|
| /ś/ | 0.60 ± 0.55 | 0.43 ± 0.65 | 0.48 ± 0.75 | n.s. |
| /ć/ | 0.40 ± 0.55 | 0.57 ± 0.76 | 0.52 ± 0.81 | n.s. |
| /ź/ | 0.60 ± 0.55 | 0.57 ± 0.76 | 0.57 ± 0.81 | n.s. |
| /dź/ | 0.60 ± 0.55 | 0.50 ± 0.76 | 0.57 ± 0.81 | n.s. |
| /s/ | 0.00 ± 0.00 | 0.00 ± 0.00 | 0.00 ± 0.00 | n.s. |
| /z/ | 0.00 ± 0.00 | 0.00 ± 0.00 | 0.00 ± 0.00 | n.s. |
| /c/ | 0.00 ± 0.00 | 0.00 ± 0.00 | 0.00 ± 0.00 | n.s. |
| /dz/ | 0.00 ± 0.00 | 0.00 ± 0.00 | 0.00 ± 0.00 | n.s. |
| /sz/ | 1.00 ± 0.71 | 0.50 ± 0.52 | 0.33 ± 0.66 | n.s. |
| /ż/ | 1.20 ± 0.84 | 0.36 ± 0.50 | 0.45 ± 0.68 | Mild > Moderate Mild > Severe |
| /cz/ | 1.00 ± 1.00 | 0.29 ± 0.47 | 0.33 ± 0.66 | n.s. |
| /dż/ | 1.00 ± 1.00 | 0.29 ± 0.47 | 0.33 ± 0.66 | n.s. |
| /r/ | 1.60 ± 1.14 | 0.36 ± 0.75 | 0.76 ± 1.34 | n.s. |
| /l/ | 0.00 ± 0.71 | 0.43 ± 0.85 | 0.14 ± 0.35 | n.s. |
| /f/ | 0.00 ± 0.00 | 0.00 ± 0.00 | 0.00 ± 0.00 | n.s. |
| /w/ | 0.00 ± 0.00 | 0.00 ± 0.00 | 0.00 ± 0.00 | n.s. |
| /t/ | 1.20 ± 0.45 | 0.36 ± 0.63 | 1.05 ± 1.07 | Severe > Moderate |
| /d/ | 1.20 ± 0.45 | 0.36 ± 0.74 | 0.90 ± 1.00 | n.s. |
| /n/ | 1.20 ± 0.45 | 0.43 ± 0.76 | 0.76 ± 1.04 | n.s. |

Analysis of variance (ANOVA) performed. n.s.—not statistically significant.

## 4. Discussion

The presented study proved that ankyloglossia is, in most cases, a condition which needs to be treated surgically, as it may cause developmental consequences for the maxillo-facial skeleton and function. The group consisted of adults who did not have frenotomy, frenectomy, or Z-plasty in childhood. As Bin-Nun [10] stated, during the last 60 years, there has been a dramatic increase in TT diagnoses, which may indicate why the tested group did not get proper treatment in the past. Zaghi et al. [14] presented a new way of assessing posterior tongue mobility using lingual palatal suction, which may be an answer for those researchers who overdiagnose TT. Moreover, it has been suggested that the mandible grows to a predetermined size and shape accompanied by some extrinsic factors. Proffit [15], in his theory on the influence on teeth positioning, stated that the teeth and the alveolar ridge lie in some balance position between cheeks, lips, and tongue, while the tongue is the most powerful, and in this way, cause the malocclusion. This research proved a relationship between ankyloglossia and laterogenia and, in general, any types of prognathism, similar to Hopkin [16] and Withman and Rankow [17], who indicated that most Class III patients had genioglossus muscle fibers that are too short or have an abnormal position. Moreover, Hopkin [16], who studied the tongue posture, pointed out that the tongue position was the highest in Class II, the lowest in Class III, and intermediate in Class I malocclusions. In addition to that, Defabianis [18] pointed out that the equilibrium between bones and muscles is vital to keep the arch widths, and an incorrect tongue's position may cause the wrong mandible position.

Another significant result is the restriction of tongue mobility in all directions. Such condition may affect the primary functions such as swallowing (here may be a particular link, however, due to a small sample, the study did not bring to light a strong relationship), chewing, cleaning the oral cavity of food, improper breathing patterns, and the tongue's most crucial resting position. As seen, the vertical–horizontal position of the tongue, known in logopaedics as resting position, was much better in those patients who had undergone the frenotomy. All upward movements improved, as well as the total tongue efficiency.

A statement about a relationship with tongue thrust may be found in plenty of articles [9,11,15,16,18]. However, the present study did not prove any relationship due to the small sample of patients with tongue thrust and the coexisting TT. It also did not verify any relation between the abnormal size of tonsils and adenoids and restricted mobility of the tongue or any indication that these factors may be relevant with ankyloglossia.

As far as speech is concerned, the results show that tongue-tie is one of the most critical factors of phonemes' quality of pronunciation across most languages, followed by a malocclusion. Several researchers such as Profitt [15], Belmehdi [19], Webb [20], Kotlow [2], Lalakea and Messner [21], proved that the short lingual frenum seriously affects speech. However, most of them covered childhood or puberty problems and did not follow the state of adults.

This research focused on the quality of speech of Polish adults in whom TT coexisted with different malocclusion of varying severity. Generally, worldwide literature points out that alveolar plosives, palatal, dental, or trill phonemes are deformed, missed, or mispronounced most frequently. In the presented paper, 19 phonemes were examined. They were placed in two groups of international and local phonemes. However, there was a third group called all phonemes. The results show that consonants like /t/, /d/, /n/ are very prone to any frenulum length or tongue's mobility changes in most languages. The inquisitive thing was Polish R's result, in which a too-short frenulum mainly causes wrong articulation. There was not any significant influence. However, patients after frenotomy pronounced the trill consonant better, which was related to less limited mobility of the tip of the tongue. Similar results concern phoneme L, which needs an excellent upward movement to the palate to be pronounced perfectly [11].

There was a significant improvement in articulating dental, palatal, and medioalveolar sounds in the local phonemes group. As most of them do not exist in Germanic languages, there is a shortage of research to compare. Some of the medioalveolar sounds in Polish are pronounced with palatalization, which means that the tongue requires upward movement and constriction, for which a proper placement of tongue and teeth is essential. If the tongue is misplaced due to a too-short frenulum and a severe malocclusion, such as any type of prognathism or skeletal open bite, those phonemes have different acoustics and are considered incorrect. Those findings agree with Belmehdi [19] and Stańczyk et al. [22]. What can be added is the fact that some compensation during the speech is present. Stańczyk pointed out that the mechanism of compensation depends on the mouth's opening width while pronouncing, the tongue touches the palate, which may cause the speech to be unclear. Another issue may be a changed tongue position in those phonemes which require lifting the whole tongue or the tip, and finally some asymmetric placement of the tongue, which in most cases results in sound deformation. The same author also adds that some increase in lateral and protrusive tongue movement may be observed in ankyloglossia, often accompanied by intensive salivation. In logopedics, dentistry, or orthodontics, every compensation can cause fatigue, occlusal discomfort, and even be a cause of dental or skeletal malocclusion.

## 5. Conclusions

Tongue-tie is a severe condition, especially in adults, leading to a permanent effect on speech. Implementing surgical procedures to treat it improves the tongue's mobility in every direction and positively impacts speech clarity. If the frenulum is too short, it causes dental and orthodontic problems and lessens the adults' effort to speak correctly, however not in every case. Swallowing patterns in patients with TT should be carefully examined in the future.

The frenotomy conducted in adults assisted excellently in recovering from problems mentioned above and may prevent other too-short frenulum consequences. Moreover, it helps with improving primary functions such as mastication, deglutition, and chewing. The successful frenotomy and therapy positively impact patients' quality of life.

The frenotomy should be implemented in patients suffering from malocclusion because the equilibrium of all the face muscles and skeleton is often disturbed and may lead to less stable effects of orthodontic and orthognathic treatment, as far as the function is concerned.

**Author Contributions:** Conceptualization, A.L. and M.K.; methodology, A.L.; software, M.K.; validation, A.L. and M.K.; formal analysis, M.K.; investigation, A.L.; resources, A.L.; data curation, A.L. and M.K.; writing—original draft preparation, A.L.; writing—review and editing, A.L. and M.K.; visualization, M.K.; supervision, M.K.; funding acquisition, M.K. All authors have read and agreed to the published version of the manuscript.

**Funding:** This research was funded by the Medical University of Lodz grant numbers: 503-1-138-01-503-51-001-17, 503-1-138-01-503-51-001-18, and 503-1-138-01-503-51-001-19-00.

**Institutional Review Board Statement:** The study was conducted according to the guidelines of the Declaration of Helsinki and approved by the Institutional Ethics Committee of the Medical University of Lodz (protocol code RNN/73/19/KB approval date: 12 February 2019).

**Informed Consent Statement:** Informed consent was obtained from all subjects involved in the study.

**Data Availability Statement:** Not applicable.

**Conflicts of Interest:** The authors declare no conflict of interest.

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
