# Peer review of "The Effectiveness of Frenotomy on Speech in Adults"

_applsci, doi:10.3390/app11062727_

Round 1

Reviewer 1 Report

RE: applsci-1155245

The effectiveness of frenotomy on speech in adults

This article showed the effectiveness of frenotomy on pronunciation in adults. This study is very meaningful because it is important to improve their QOLs after surgery. There are few things to be rewritten to be better.

  1. Describe the detail of the exercise before and after the surgery (line 105).
  2. Describe who evaluated the pronunciation (line 134-138).
  3. It is important what type of patient could improve their pronunciation after surgery (mild/moderate/severe ankyloglossia).

Author Response

Dear Sirs, 

Thank you for all notes on our manuscript. 
Answering your questions we'd like to inform you that:

1.  Those exercises included: gentle massage of the mouth’s floor along with the frenulum itself (using the index finger), horizontal and vertical movements of the apex, preadorsum, mediodorsum and postdorsum. Vertical movements of mediodorsum are movements in which the middle part od the tongue moves upward to the palate and the tip is placed lose to lower central incisors. Postdorsum movements were connected with coughing and moving the soft palate. Due to such movements, patients were able to identify the location on the tongue and particular muscles after the frenotomy. Besides, they were instructed to open mouth wide so as not to feel any discomfort. Patients were asked to do some basic physical activities of the whole body like stretches facing down, neck rotations, and bends accompanied by the massage of glottal and subglottal muscles.

2. The pronunciation assessment was designed and provided by a qualified and experienced speech therapist using the adaptation of a pronunciation test for teenagers. 

3. The most significant changes in the speech were visible in patients with severe and moderate ankyloglossia who after the surgical treatment has a different type of TT from mild to moderate. We are providing another table with results as an attachment, which will be in the manuscript. 

Moreovoer, we will add more references as we got new papers which should be addressed in our paper. 

Your faithfully, 
Anna Lichnowska

Reviewer 2 Report

The purpose of this study was the functional evaluation of lingual frenotomy on tongue 19
mobility and speech in the adult Polish population

The author mentioned the approach is concerning tongue-tie (TT) in adults. Is there any difference between adults and childs?

What is the main novelty of the proposed framework?

I recommend to use most recent paper in the field of speech in related work such as:

Zaghi, S., Shamtoob, S., Peterson, C., Christianson, L., Valcu‐Pinkerton, S., Peeran, Z., Fung, B., Kwok‐keung Ng, D., Jagomagi, T., Archambault, N. and O’Connor, B., Assessment of posterior tongue mobility using lingual‐palatal suction: progress toward a functional definition of ankyloglossia. Journal of Oral Rehabilitation.

Gogate, M., Dashtipour, K., Adeel, A. and Hussain, A., 2020. CochleaNet: A robust language-independent audio-visual model for real-time speech enhancement. Information Fusion, 63, pp.273-285.

I would recommend as future work use machine learning and deep learning to detect the tongue-tie (TT) 

Author Response

Dear Sirs, 

Thank you so much for all the notes on our manuscript.

We are grateful for another paper to our manuscript. We will for sure address it in our work. We know this technique and we are also examining it for a different study. 

Answering your questions we would like to inform that the main novelty about the framework is the holistic approach to adults and their combined treatment of skeletal deformities by surgeons and orthodontist who are supported by speech therapists and physiotherapists. Due to such an approach, we are able to diagnose multiple things at one time, moreover, the presence of speech therapists in a team with MDs helps to stabilize the effects of all orthognathic surgeries as they are able to detect tongue thrust and disorders concerning biting or chewing patterns which during a long term may be unprofitable for patients, especially when those disorders concern swallowing. Foletti  et(2017) pointed out that the stability of orthognathic surgery is based on the functioning of the muscles and the tongue is a muscle, a very strong one. Foletti pointed out that the main risk factor concerns malocclusion from II and III Class. 

2. The main difference between children and adults is a compensation mechanism, mainly absent in children unless the TT is severe or completed. The steps in our protocol are different as our patients had to exercise for a longer period of time prior to and after the surgical treatment. Moreover, the pace of change of the patterns of primary functions such as mastication, swallowing, biting, and chewing differs between children and adults. 

We are attaching new results which will be added to our work comparing the types of frenotomy and progress in pronunciation. 

Yours faithfully, 

Anna Lichnowska
Anna Lichnowska
